# A Pilot Feasibility Study of an Intensive Summer Day Camp Intervention for Children with Selective Mutism

**DOI:** 10.3390/children9111732

**Published:** 2022-11-11

**Authors:** Danielle Haggerty, John S. Carlson, Aimee Kotrba

**Affiliations:** 1Children & Adolescent Behavioral Health, McCaskill Family Services, Brighton, MI 48114, USA; 2Department of Counseling, Educational Psychology & Special Education, Michigan State University, East Lansing, MI 48824, USA; 3Children & Adolescent Behavioral Health, Thriving Minds Behavioral Health and Family Services, Brighton, MI 48116, USA

**Keywords:** selective mutism, intensive intervention, behavioral therapy, anxiety, summer camp, children

## Abstract

Cost, scheduling, and implementation competency are barriers to accessing traditional evidence-based behavioral interventions for childhood selective mutism (SM). Brief, or intensive, interventions are a disruptive innovation to traditional therapy given the use of fewer sessions during a short-term time period. This study explored the acceptability, integrity, and effectiveness (i.e., single-case replicated AB design) of an intensive summer camp consisting of a 5-day behavioral therapy for 25 children with SM. Caregiver-rated treatment acceptability ratings and family interviews support intensive summer day camp as an acceptable intervention approach for SM. Additionally, results revealed that counselors and parents implemented SM behavioral therapy during camp with impressive integrity (>90%) after receiving training about SM behavioral therapy from an SM expert clinician. Effect size calculations of counselor-rated daily behavior ratings revealed reductions in anxiety during camp for 18 of the 25 campers. Significant caregiver-rated improvements in speaking behaviors were reported for 9 out of 14 campers with data available for analysis at the 3-month follow-up. This pilot feasibility study is the first to investigate intensive summer day camp as a treatment approach for SM and implications for future research are discussed.

## 1. Introduction

Selective mutism (SM) is a rare anxiety disorder characterized by the consistent inability to speak in settings where the expectation is to initiate speech or engage in reciprocal communication [1]. The prevalence of SM is estimated to be less than 2%, with an onset age of before five. Children with SM may speak comfortably in certain situations, particularly at home, and persistently fail to speak in other social settings (e.g., school, community).

SM is classified in the Diagnostic and Statistical Manual for Mental Disorders as an anxiety disorder, which aligns with scholars’ consensus of close etiology links between SM and social anxiety disorder (SAD), specifically [1]. Beyond etiology, behavior inhibition to escape stress inducing situations is a common coping strategy for individuals with anxiety [2]. As with other anxiety disorders, SM often leads to dysfunction in a child’s life and warrants treatment [3]. Children with SM engage in a pattern of avoidance to decrease feelings of anxiety in environments where speech is expected [4]. The maladaptive cycle starts with a prompt for response, feelings of anxiety, avoidance of speech (e.g., parent speaks for child), and then decreased anxiety. Escaping aversive physiological and psychological effects of anxiety by withholding speech negatively reinforces the behavior and thus, the child is likely to avoid speaking in the future. Furthermore, parents/caregivers are reinforced for rescuing behaviors, and are therefore more likely to rescue via speaking for the child in the future. Lack of speech can result in social isolation and academic and social dysfunction, which means the behavioral avoidance cycle needs to be disrupted through exposure-based treatment.

Given the conceptualization that withholding speech fulfills a behavioral function for children with SM, behavioral therapy is a commonly recommended treatment for the disorder [5]. Typical behavioral therapy for children with SM includes contingency management and shaping strategies, which include direct instruction and positive reinforcement for target behaviors. Stimulus fading, role-playing, and hierarchal exposure are common supplements to typical therapy as it builds on the child’s success with exposure to speaking in an increasing number of environments. The goal of behavioral treatment is for the child to experience decreased symptoms of SM (i.e., speech in several environments) as they increase skills and confidence with progressively more difficult exposure tasks.

In a recent analysis of peer-reviewed journal articles about SM treatment between 2005 and 2015, behavioral therapy appeared to be the most popular and effective therapy for SM [6]. Behavioral therapy, in conjunction with another therapy (e.g., psychodynamic, systems), was implemented in 21 out of 23 articles reviewed and was the exclusive therapy used in seven of those articles. Six out of seven studies included only one or two participants. In the additional study, three participants were not included in the total sample size (*n* = 9) because they dropped out after three sessions [7]. Participants from the seven studies were between three and 13 years old and included 12 females and five males who were European American, African American, Asian American, Hispanic/Latina, and biracial. Therapy was implemented in a variety of settings, including school, clinic, and the community. All seven studies found improvements in SM symptoms, including increased responses, initiations of speech, verbalizations, and improved teacher and parent rating scale scores following a range of one to seven months of behavioral therapy.

While behavioral therapy appears to be a common and effective intervention for children with SM, there are limitations to its implementation in a traditional therapy format. Accessibility is a limitation to traditional behavioral therapy. Experts in treating SM are scarce, due in part to the rarity of the disorder [4]. Second, families may encounter barriers seeking treatment due to costly 11 to 18 weekly sessions [8]. Weekly treatment can cost between $100–$200 per hour, not including weekly travel costs (e.g., gas, bus, train, tolls). Third, scheduling challenges over the traditional therapy duration of three to six months may result in lost momentum with treatment effects. Fourth, even if a family can access treatment, the expectation of school personnel to implement the clinician-designed intervention may imply gaps in competency and execution of intervention plan [7].

Disruptive innovations may circumvent these barriers (i.e., access, cost, scheduling, implementation competency) through novel forms of delivery of evidence- based interventions (EBIs) [9]. Disruptive innovations synthesize common, robust EBI elements and aim to serve more people for lesser cost by meeting “the essential needs of the majority of consumers in more efficient and accessible ways” [9], p. 467. As a disruptive innovation, brief interventions are a novel delivery format to reach more people through efficient treatment which includes less sessions than traditional therapy. The time and financial commitments associated with brief interventions may help to overcome scheduling and cost limitations associated with traditional behavioral therapy. Brief interventions as a screener to help guide individualized treatment aligns with previous treatment recommendations given the heterogeneity of symptoms and resulting dysfunction appearing across settings seen in children diagnosed with SM [10].

Brief, Intensive, and Concentrated (BIC) interventions are an adaptation of brief interventions described above, as they share a foundational rationale to address the limitations of traditional therapy through short-term intervention but often have more than one or two sessions. A recent meta-analysis defined “brief” as interventions with notably less sessions than traditional therapy, “concentrated” as interventions with more than one session per week in short time period, and “intensive” as interventions that were both brief and concentrated [11]. This meta-analysis provides initial support for BIC cognitive behavioral treatments (CBT) for anxiety disorders in children. A total of 23 randomized clinical controlled trials (RCTs) were reviewed, including 13 studies about specific phobias, three studies about obsessive- compulsive disorder (OCD), three studies about posttraumatic stress disorder (PTSD), and one study each about panic disorder, separation anxiety disorder, SAD, and mixed anxiety disorder. The format of therapy was individual for 20 studies and group for three studies. Results suggest strong acceptability of BICs, as evidenced by the 6% decline rate for family participation in the RCTs and 2% dropout rate once intervention had begun. Intervention integrity was not reported in this meta-analysis. Results showed a very large effect size (g = 1.47) when BICs were compared to waitlist controls, a large effect size (g = 0.97) when BICs were compared to placebo conditions, and no difference (g = 0.01) when BICs were compared to traditional (i.e., once per week for 11 to 18 weeks) CBT. This meta-analysis provides promising results to suggest BIC interventions have the same results of traditional CBT, while addressing its barriers, especially those related to access. Given the clarity and comprehensiveness of the definition of “intensive” intervention to include “brief” and “concentrated” approaches [11], this current study’s intervention approach is referred to as “intensive” hereafter.

While there is well-developed evidence to support the implementation of RCTs for intensive interventions for anxiety disorders such as specific phobias, there is less information available about intensive interventions for SM and closely related disorders (e.g., SAD). Specifically, the seven behavioral therapy-exclusive articles included the SM intervention meta-analysis [6] would not fit the criteria of BICs because treatment duration extended one to four months. Literature about SM treatment approaches are dominated by case studies [6] and lacks exploration of intensive interventions as a treatment approach to SM or closely related disorders (e.g., SAD). Feasibility studies are warranted when “there are few previously published studies or existing data using a specific intervention technique” [12], p. 453. Acceptability, implementation (i.e., integrity), and limited-efficacy testing (i.e., effectiveness) are essential areas of study when investigating BIC interventions for anxiety disorders [12]. Acceptability is defined as satisfaction or the extent to which the intervention is suitable and satisfying to the consumer. Implementation is defined as integrity or the extent to which the intervention was executed as planned. Limited efficacy is defined as the exploration of whether the intended effects of the intervention occurred and the consideration of intervention effectiveness in a future study with more controlled conditions (e.g., RCT).

Acceptability, integrity, and effectiveness are fundamentally linked [12]. An original factor analysis of the Treatment Evaluation Inventory TEI [13] revealed acceptability and effectiveness load on the same factor for the TEI, which measures parent intervention acceptability. Meanwhile, previous literature shows a positive correlation link between acceptability and integrity, which suggests consumers are more likely to implement intervention with integrity when they consider it highly acceptable [14]. The distinct connection between these variables make it essential to measure all three when assessing an intervention, like BIC.

Treatment acceptability is infrequently reported in the literature exploring intensive intervention approaches for individuals with anxiety disorders. Three studies of varying sample sizes (*n* = 3, *n* = 23, *n* = 40), participant ages (7–18) and anxiety disorders (social phobia, SAD, OCD) are compared. One study included 40 children (ages 7–12) with social phobia, divided into treatment and control waitlist groups [8]. Treatment participants received four 3-h sessions, including psychoeducation content and behavioral exposures, over the span of three weekends (15 days) in groups of four to six children. It was mentioned anecdotally that the intervention was highly acceptable to parents and children, as well as revealed through an average score of 3.83 (higher than “quite a bit satisfied”) on an eight-item 5-point Likert scale for acceptability. A second study involved three adolescents (ages 13–18) who received ten 1-h CBT sessions with the therapist and their parent over five days [15]. All three participants agreed the treatment was rigorous but useful. Though parent and child anecdotes of acceptability are promising, information from more reliable measures is needed to understand which components of intensive intervention lend to its acceptability. The third study included 23 children (ages 8–11) with social phobia randomly divided into a treatment and waitlist control group [16]. Treatment participants received three 3-h sessions, including psychoeducation content and behavioral exposures, over the span of three weeks in groups of five to seven children. No mention of treatment acceptability was made.

The integrity of intensive intervention implementation is also infrequently and not well documented in the literature when it is used as a novel delivery format for anxiety disorders. Two of the three studies previously reviewed [5,8]. The third study included videotapes of all sessions used to determine the adherence to the treatment protocol and noted adherence was reviewed and addressed after the first session to improve integrity [16]. However, their review of treatment adherence was vague, and rate of adherence was not reported.

The outcomes of these studies provide initial support for intensive interventions as effective interventions for anxiety disorders like SM. For example, it was found that by posttreatment, 52.4% of treatment participants did not meet diagnostic criteria for social phobia, compared to 15.8% of controls [8]. By 6-month follow-up, 76.9% of treatment participants did not meet diagnostic criteria for social phobia. At posttreatment and 6-month follow-up, treatment participants reported a greater improvement in anxiety symptoms, internalizing problems, depression, social skills, social competence, and parental social anxiety symptoms than control participants. In a different study, OCD symptoms decreased and functioning increased for all three participants, while two participants experienced a 40% decrease in symptoms [15]. The additional study found that at posttreatment, diagnostic interviews, parent reports, and child reports indicated treatment group participants showed significantly improved social phobia related symptoms compared to controls [16].

A ten-step intervention research trajectory for the development and testing of intervention approaches has been proposed [17]. In this progress model, after identification of an issue (i.e., SM diagnosis) and strategies (i.e., behavioral therapy through intensive intervention) to address the issue, strategies should be tested for feasibility via a pilot study. The testing of behavioral therapy as an intervention for SM has been well documented in the SM literature with positive outcomes [5,6,18]. However, none of the SM treatment studies examined the use of behavioral intervention in an intensive format. Thus, according to the research trajectory [17], it is appropriate to implement a pilot study to assess the feasibility of an intensive intervention for children with SM. Specifically, piloting an intensive intervention in non-traditional settings like summer camps create “the potential for an intensive therapeutic experience, coupled with the innovation of a camp setting that includes fun, reinforcing activities…fit well within a day camp, short-term treatment model” [19], p. 360.

One feasibility study to explore intensive summer camp intervention for children with separation anxiety disorder has been conducted [20]. Specifically, the feasibility of intensive CBT for five girls (ages 8–11) with separation anxiety disorder during a seven-day summer camp was investigated. A case-series design including baseline, pretreatment, posttreatment, and three-month follow-up assessment measures indicated high rates of parental acceptability; parents were “very satisfied” with their daughters’ progress. Intervention integrity was not reported for this study. All participants experienced significant decreases in separation anxiety disorder symptoms at posttreatment, and none of the participants met diagnostic criteria at three-month follow-up.

### Current Study

The present study aimed to parallel the prior separation anxiety disorder study [20] by piloting an intensive summer day camp intervention for children with SM, called Confident Kids Camp (CKC). The current study utilized a non-randomized replicated AB single-case design to examine the acceptability, integrity, and effectiveness of intensive intervention implemented in a 5-consecutive day summer camp for 25 children with SM. It extends prior research on SM and it builds in additional methodological rigor by doing (a) family interviews to explore the acceptability of time, resources, and accessibility of this 5-day camp, (b) integrity checklists and integrity observations, (c) daily tracking of child-level anxiety levels and speaking behaviors including video recording, and (d) replication across participants. These study components increased the rigor of the prior intensive intervention studies previously described.

## 2. Materials and Methods

### 2.1. Participants

Twenty-five campers participated in CKC Summer 2019, consisting of 22 families, given three sets of siblings. Fourteen campers traveled from nine different states (Michigan, Ohio, Indiana, Wisconsin, Iowa, Minnesota, California, Hawaii, Arkansas) and Canada. Twenty of the 25 campers were female and five campers were male. Twenty-two campers were white. Campers were separated into three grade-based classrooms. The younger class included pre-kindergarten to kindergarten-aged campers (ages 4–5; *n* = 7), the middle class included first through third grade aged campers (ages 6–8; *n* = 9), and the older class included fourth through sixth grade aged campers (ages 8–11; *n* = 9). Ages of campers ranged from four to eleven (*M* = 7 years, 11 months) at the start of camp.

#### Inclusion and Exclusion Criteria

Eligible campers were between the ages of four and 12 and had a primary diagnosis of SM. Children could present with a range of SM severity, though children were not eligible for participation in camp if they could not establish speech with their assigned counselor during the lead-in session prior to the start of camp. Given the primary purpose of the feasibility study being the exploration of hypotheses to move the SM intensive intervention literature to more rigorous designs, exclusion criteria were limited. Children exposed to previous treatment participated in camp. Zero children with comorbid diagnoses of neurodevelopmental disorders, major mood disorders, or psychotic disorders enrolled in camp. Ultimately, all children enrolled in camp participated and their data were extracted for this study.

### 2.2. Treatment Acceptability Measures

#### 2.2.1. Treatment Evaluation Questionnaire-Parent Form

All caregivers completed the Treatment Evaluation Questionnaire-Parent Form TEQ-P [13] at posttreatment to rate their level of acceptability of the treatment. The TEQ-P was organized by three subscales, including Acceptability, Effectiveness, and Time Required. Caregivers rated their experiences of acceptability and intervention quality on a Likert Scale from 0 (strongly disagree) to 6 (strongly agree), and higher scores indicated higher levels of acceptability. Scores could range from 21 to 126. An overall score of 110 indicates high levels of caregiver acceptability, as this score was the sum of high-level scores determined from each subscale [21]. Specifically, subscale scores at or above 55, 36, and 9 for the Acceptability, Effectiveness, and Time Required subscales, respectively, indicate high levels of acceptability. The TEI [22] items, from which the TEQ-P was adapted, had high internal consistency (α = 0.97). Overall scores were averaged for all caregivers.

#### 2.2.2. Family Interviews

Family interviews of acceptability were conducted at posttreatment. The interview questions were derived to align with (a) the three TEQ-P subscales, (b) identified item factor loading in the TEI [13], and (c) identified barriers of traditional treatment that intensive intervention aims to circumvent. Interview questions probed the acceptability of the intensive summer day camp intervention and, when applicable, explored whether it addressed the barriers associated with traditional therapy (e.g., “How did the cost of CKC compare to previous treatment?”). While there was no reliability or validity information available for the interview, the questions were strategically derived from (a) TEQ-P which has high internal consistency (α = 0.97) and barriers identified in research [9]. Interviews were coded for themes.

### 2.3. Treatment Integrity Measures

#### 2.3.1. Counselor Implementation Integrity

Counselors completed integrity rating scales every day immediately after camp for their implementation efforts with their assigned camper. Integrity checklists were developed based on the CKC curriculum, and reflected each day’s and classes’ activities with corresponding ratings on a 4-point scale of implementation, with a range from 0 (Not observed), 1 (Implemented inappropriately), 2 (Implemented somewhat appropriately), to 3 (Implemented appropriately). Items were coded dichotomously as Yes or No, with scores from 0 to 1 being coded as No and scores from 2 to 3 being coded as Yes. Integrity rates were calculated by dividing the number of Yes codes by the number of possible activities. The minimum rate for adequate treatment adherence is 80% or higher [23].

Inter-rater reliability checklists were completed by classroom teachers and the camp director for 14 counselors, and their results were generalized for counselors’ implementation integrity. The 14 counselors were chosen because their campers were covered by an insurance provider that required one-hour observation from a licensed professional. Measuring inter-rater reliability addressed the lack of reliability and validity information available about this integrity check tool. Inter-rater agreement was calculated using percentages of agreement between teacher and counselor integrity checklists. The strength of inter-rater agreement for categorical data is “almost perfect” with a kappa statistic of 0.81 [24].

#### 2.3.2. Caregiver Implementation Integrity

An integrity checklist was developed to measure caregiver implementation integrity of intensive summer day camp intervention techniques during the community-based exposure camp activity. The caregiver integrity checklist used the same scale and scoring as the counselor integrity checklist. The counselor completed the integrity checklist for their assigned camper’s caregiver(s) immediately following the community exposure activity, and the caregiver rated their implementation efforts on the same scale. Inter-rater agreement was calculated using percentages of agreement between counselor and caregiver integrity checklists, and the goal of inter-rater reliability was a kappa statistic of 0.81 [24].

Additionally, caregivers completed a very similar integrity checklist throughout the 3-month period between posttreatment and follow-up to measure caregiver behavior change. For each planned exposure activity during that time, caregivers completed the integrity checklist to determine their implementation integrity with facilitating exposures for their child. Caregiver integrity checklists were used to determine treatment integrity percentages, with the goal to implement exposure activities with at least 80% adherence over time [23], 2005.

### 2.4. Treatment Effectiveness Measures

#### 2.4.1. Anxiety Levels

The Screen for Child Anxiety Related Disorder, Parent about Child Version SCARED [25] and a Daily Behavior Report (DBR) were used to measure camper (ages eight or older) anxiety levels, as reductions in anxiety are the hypothesized mechanism of change for increased speaking behaviors. A parent version was completed by caregivers at pretreatment, posttreatment, and three-month follow-up. Both SCARED measures are brief broadband 41-item measures. Caregivers rated the frequency of their child’s anxious behaviors on a Likert Scale from 0 (not true or hardly ever true), 1 (somewhat true or sometimes true), or 2 (very true or often true), with higher scores indicating more frequent anxiety symptoms. At pretreatment and three-month follow-up, caregivers completed the survey based on their child’s behavior for the past three months. At posttreatment, caregivers were asked to complete the survey based on their child’s behavior over the past week, in order to assess for anxiety level changes during camp. The child measure is identical to the parent measure for children (i.e., Likert scale and time frame) with the adjustment of framing questions in the first person. The SCARED is organized by five subscales, including Panic Disorder/Significant Somatic Symptoms, Generalized Anxiety Disorder, Separation Anxiety, Social Anxiety, and Significant School Avoidance. Scores range from 0 to 82, with scores over 25 indicating further examination for the presence of an anxiety disorder. A reliability of α = 0.90 for the SCARED parent and child report have been reported [25], indicating high internal consistency for all items. The SCARED was analyzed using Reliability Change Index (RCI) for each participant from pretreatment to posttreatment and from posttreatment to three-month follow-up. RCIs were interpreted as effect sizes and guidelines, with RCIs below −1.8, indicating significant reductions in anxiety scores [26].

A Daily Behavior Report (DBR) was individually formed for each camper and included the three highest-scored items from the highest-scored SCARED subscale, as indicated by parent report at intake. DBRs are an evidenced-based tool to track child behavior and include operationalized definitions of target behaviors and specific criteria for the child meeting behavioral goals [27]. Aligned with the promotion of including a heterogeneous sample for this study, DBRs reflected the foundational anxiety symptoms of each participant, as it is hypothesized repeated practice of an alternative behavior (i.e., speech) would demonstrate decreases in these symptom scores.

On the DBR, counselors rated the frequency of their camper’s anxious behaviors for five exposure activities each day on the identical SCARED Likert Scale from 0 (not true or hardly ever true), 1 (somewhat true or sometimes true), or 2 (very true or often true), with higher scores indicating more frequent anxiety symptoms. Recent meta-analytic results [28] show high coefficient alphas for each of the five SCARED subscales, which include Panic Disorder/Significant Somatic Symptoms (α = 0.84), Generalized Anxiety Disorder (α = 0.81), Separation Anxiety (α = 0.72), Social Anxiety (α = 0.78), and Significant School Avoidance (α = 0.62). Additionally, significant correlations (0.81) between mean teacher DBR ratings compared to systematic observations exist, which suggest DBRs are a reliable supplement to more established measures of behavior like the SCARED [29]. Anxiety level change, as measured by the DBRs, was analyzed using daily averages, visual analyses (effect, and consistency of patterns across cases), and effect size calculations using the Interrupted Time-Series Simulation (ITSSIM) computer software [30]. A possible range of mean scores was produced during the simulation process in ITSSIM and effect sizes were produced with larger effect sizes indicating less overlap between predicted A and B phase points.

#### 2.4.2. Speaking Behaviors

Caregivers completed the Selective Mutism Questionnaire (SMQ) [31] at pretreatment, posttreatment, and three-month follow-up. The SMQ is a 17-item measure and provides a clinical profile of the child’s SM by assessing their speech inhibition across settings, specifically at school, at home/family, and in social situations outside of school. Caregivers rated the frequency of their child’s speaking behaviors on a Likert Scale from 0 (never), 1 (seldom), 2 (often), to 3 (always), with lower scores indicating lower frequency speaking behaviors. Scores could range from 0 to 51, with a mean score of 12.99 (*SD =* 7.23) for children with a primary diagnosis of SM [30]. While cutoff scores for the SMQ are unavailable, the mean score of approximately 13 is often referenced to determine SM severity. The internal consistency of SMQ factors (α between 0.654 and 0.913) and total scores (α = 0.783) ranged from moderate to high [32]. The convergent and incremental validity were also found to be strong. The SMQ was analyzed using Reliability Change Index (RCI) for each participant from pretreatment to posttreatment and from posttreatment to three-month follow-up. RCIs were interpreted as effect sizes and guidelines, with RCIs above 1.8 indicating significant improvements in speaking behavior [26].

#### 2.4.3. Observed Speaking Behaviors

As a part of single-case design, it was critical to track the target behavior of change (i.e., speech avoidance) to adequately assess whether the mechanism of change occurred simultaneously to behavior change. Video recordings of camper speaking behaviors during five daily exposure activities were coded to track changes across baseline and intervention phases for all 25 campers. During five consistent exposure activities each day (i.e., PRIDE, morning meeting, novel exposure, prize store, and closing assembly), rate of camper responsive and spontaneous speech was coded by quantifying the total number of responsive and spontaneous words spoken and dividing those numbers by total number of minutes for each camper. Responsive and spontaneous speech rates were aggregated, separately, for each day due to different speech demands for each exposure but similarity of exposures from day to day. Given there was not reliability or validity information for this tracking mechanism, audio-recording and video-recording of each exposure was used to maintain objectivity. Additionally, visual analyses (effect, and consistency of patterns across cases) and effect size calculations using the ITSSIM computer software [30] were used to analyze speaking behavior change.

### 2.5. Procedures

#### Project Personnel and Training

The camp director is the lead CKC clinician and a licensed clinical psychologist with an expertise in behavioral treatment of SM. The lead CKC clinician coordinated CKC: camp enrollment, developing and coordinating the CKC curriculum, providing training to teachers and counselors before treatment, providing parent training for two hours daily during CKC, and providing leadership guidance and supervision during the camp. The classroom teachers were three clinicians with experience treating SM. One teacher was a licensed psychologist and the other two teachers obtained a Temporary Limited License (i.e., psychology intern and practicum student) in psychology. Classroom teachers’ main responsibilities included leading class activities (e.g., circle time, psychoeducation), scaffolding counselor implementation, and role-playing as novel adults or school-based classroom teachers for exposure practices.

Camp counselors were graduate students in a psychology-related field or school-based employees in a related field (e.g., teacher, school psychologist, social worker, speech-language pathologist). All first-time counselors attended a one-day training a couple of months before CKC, led by the lead CKC clinician. Training consisted of an explanation of SM, behavioral treatment approaches to break the avoidance cycle, overview of CKC camp, and role-playing to practice exposure activities with campers.

### 2.6. Treatment Phases

#### 2.6.1. Pretreatment Phase

Acceptability was determined at pretreatment as child enrollment in camp between January 2019 and May 2019. Caregiver SMQ and caregiver and child SCARED rating scales were completed as part of the intake during the enrollment period. In the months leading up to CKC, speech was established between the camper and the assigned counselor during the lead-in session through fading and transferring speech strategies. An SMQ and SCARED were also completed by caregivers and children, when applicable, on the morning of the first day of camp to account for (a) the possible effects the lead-in session had on camper behavior when CKC started and (b) the duration of time between intake and the start of camp in August 2019.

Given the long span of time during which intakes could be completed (i.e., 5 months), the first day of camp rating scales were used as “pretreatment” scores. No significant differences were found between SMQ scores at intake to the first day of camp. Three camper’s caregivers scored their speaking behaviors as severe (SMQ Total score < 13) at pretreatment. The average score among the 25 campers on the SMQ was 21.64 (*SD* = 7.92, Range: 3–39). Significant differences (*p* < 0.01) were found between caregiver-rated SCARED measures from intake to the first day of camp, indicating significantly higher scores of anxiety on the first day of camp. However, each camper’s highest-scored SCARED subscale remained the same from intake to the first day of camp. Twenty camper’s caregivers endorsed social anxiety as the highest subtype of anxiety on the intake SCARED measure, while five camper’s caregivers endorsed Generalized Anxiety as the highest subtype. The most recent SMQ and SCARED were used as pretreatment scores for RCI calculations, while the intake session SCARED was used to identify DBR items. Twenty-one campers had previously received treatment for SM, while four campers had not.

#### 2.6.2. Baseline Phase (A)

Baseline speaking behavior was established for each camper during the first 20 min of free play on the first morning of camp before intervention was implemented. During this time, video tapings were coded for number of responsive and spontaneous words spoken in each four-minute increment (total of five baseline data points). Each total number of words for each four-minute increment was divided by four to equal each camper’s rate of speech.

#### 2.6.3. Treatment Phase (B)

While their children were at camp, caregivers attended a daily two-hour parent training session led by the lead CKC clinician, during which they were provided psychoeducation about SM, trained on behavioral interventions (i.e., stimulus fading and shaping), discussed appropriate educational supports for children with SM, and were encouraged to connect with other caregivers with children with SM.

Counselors and teachers implemented intervention (B) from 9:20 am until 3:00 pm on Monday through Friday (Table 1). Behavioral therapy was utilized throughout the day, especially contingency management, as counselors gave stickers or “brave bucks” (i.e., fake money) to campers every time they spoke. Campers turned in their sticker sheets and brave bucks for a prize at the end of every day.

Counselors implemented a hierarchy for eliciting communication from their camper during intervention. They progressed from verbal sound prompts, to yes/no questions, to forced-choice questions, and finally to open-ended questions to elicit speech from their camper. Additionally, teachers provided in vivo support to ensure counselors implemented the communication ladder appropriately and intervened if the camper was having an especially hard time speaking. Additionally, counselors planned for each exposure activity through goal-setting and practicing.

First, each day had a psychoeducation component that was developmentally appropriate. The younger class read books about feeling worried while middle and older classes learned relaxation techniques. Second, each day had components of exposure to prepare campers for the upcoming school year. For example, practicing interrupting a teacher, eating lunch with peers, participating in circle time, volunteering for class jobs, and recess with peers. Third, each day had either a field trip or big activity that required goal-setting and practice beforehand. These activities included art class, therapy dogs, visiting the animal conservancy, obstacle course, scavenger hunt at stores with caregivers, and ordering snacks. Other exposure-based activities such as Person Bingo and giving/receiving compliments were included in the days, as well.

Video recordings were coded for rate of responsive and spontaneous words spoken during five daily exposure activities. Each day, PRIDE (praise, reflect, imitate, describe, and enthusiasm- techniques used to establish rapport and subsequent speech) [4], morning meeting, prize store, and closing assembly were recorded. The fifth recorded exposure was different each day (i.e., art class, animal conservancy practice, therapy dogs, obstacle course, show & tell). Given different expectations of speech during each activity, but consistency in activities from day to day, responsive and spontaneous speech were aggregated for the day, separately. At the end of every day, the counselor completed their integrity checklist and the camper’s DBR, which scored camper anxiety related to each videotaped activity. On Thursday, only, the counselor and caregiver also completed an integrity checklist for the camper’s caregiver during the community-based exposure activity.

#### 2.6.4. Posttreatment

Caregivers completed the TEQ-P, family interviews, SMQ, and SCARED to provide data to inform intervention acceptability and effectiveness at posttreatment on the last day of camp (Friday). Children aged eight and older completed the SCARED child report as a part of afternoon class time.

#### 2.6.5. Three-Month Follow-Up

At three-month follow-up, the lead CKC clinician prompted caregivers to complete the SMQ and SCARED electronically, and submit caregiver-completed integrity checklists from between posttreatment and three-month follow-up. Two caregiver integrity checklists were returned, fourteen caregiver-rated SMQ and SCARED rating scales were completed, and one child-rated SCARED was completed at three-month follow-up. Three-month follow-up data analyses were run with data collected via the survey link, and missing data were not corrected for.

### 2.7. Data Analysis

Analysis of treatment acceptability was conducted with TEQ-P scores for each camper’s caregiver at posttreatment. Additionally, the posttreatment family interviews were coded for themes related to acceptability of the intensive summer day camp.

Treatment adherence was analyzed by calculating percentages of daily activities implemented by the counselors throughout CKC. Daily percentages were calculated by number of activities implemented divided by number of possible activities. Counselors’ rates of implementation adherence were calculated by averaging the percentages for all five days. The same process was used to calculate caregiver percentages. Inter-rater reliability for implementation integrity was analyzed by teachers’ and the lead CKC clinician’s supplemental integrity checks of counselors. The percentage of overlap between the counselor’s self-reported integrity and teachers’ or lead CKC clinicians’ was used to determine percent agreement.

Analysis of treatment outcomes was completed via visual analyses, effect size calculations [30], and RCI calculations [26]. Visual analyses of scores were completed for each camper by independent research assistants who used a guidebook. The guide was used as a comparison tool of agreement between the research assistants pertaining to the determination of data level, trend, variability, the immediacy of effect, and consistency of patterns across cases. It did not include an evaluation of non-overlapping data because of subsequent utilization of effect size calculations. Effect sizes were calculated using ITSSIM computer software [30]. The ITSSIM software calculated parameter estimations, the null effect and experimental effect models, “after standardizing the Theil-Sen residuals (i.e., dividing residuals by their within-phase standard deviation)” [30], p. 595. A possible range of mean scores were produced during the simulation process in ITSSIM (Tarlow & Brossart, 2018). Finally, effect sizes were produced with larger effect sizes indicating less overlap between predicted A and B phase points (Tarlow & Brossart, 2018). Separate analyses were conducted for different outcome variables. Additionally, aggregate mean effect sizes were used to explore outcome differences between CKC classroom (i.e., children age). RCI was calculated by subtracting the two scores of interest and dividing them by the outcome measure’s (i.e., SMQ and SCARED) standard error of measurement (SEM).

## 3. Results

### 3.1. Acceptability

#### 3.1.1. TEQ-P

Overall acceptability of the intensive summer day camp was not replicated across caregivers, as reported on the TEQ-P (Table 2). Six of the 25 caregivers rated adequate overall acceptability of the intensive summer day camp, and the average score (*M* = 98.2, *SD* = 11.95) did not reach the threshold score (110) for acceptability. However, the majority of caregivers (*n* = 17) endorsed adequate scores of acceptability for the intensive summer day camp’s treatment quality, and the average subscale score (*M* = 59.10, *SD* = 5.40) exceeded the threshold score (55) of acceptability. Similarly, TEQ-P results revealed the majority of caregivers (*n* = 16) endorsed adequate scores of satisfaction with the time required for the intensive summer day camp intervention, and the average subscale score (*M* = 9.48, *SD* = 2.20) exceeded the threshold score (9) of acceptability. Low caregiver scores on the effectiveness subscale negatively affected overall acceptability ratings. Specifically, six caregivers scored adequate levels of acceptability about treatment effectiveness, and the average subscale score (*M* = 29.6, *SD* = 6.73) did not reach the threshold score (36) for acceptability. Caregivers completed a TEQ-P for each camper, and no significant differences in total or subscale scores were found between (a) classes, (b) SM severity, or (c) previous treatment versus no treatment.

#### 3.1.2. Family Interviews

Results from family interviews were summarized by acceptability, time required, and effectiveness subscales (Figure 1). Family interviews were conducted with 20 out of 22 families. Two families were unable to participate in family interviews on the last day of camp due to scheduling difficulties. Both families not interviewed endorsed total scores on the TEQ-P that exceeded the threshold of acceptability (113 and 115), as well as on each subscale. Seventeen caregivers reported their child(ren) had received previous treatment for SM, including three caregivers who reported two or more previous treatment approaches (e.g., weekly therapy and previous years’ CKC) for their child(ren). Three caregivers reported their child(ren) had not previously received treatment for SM. Previous treatment included weekly to monthly clinic-based behavioral therapy (*n* = 12), weekly clinic-based play therapy (*n* = 1), clinic-based intensive behavioral therapy (*n* = 4), previous years’ CKC (*n* = 3), and a different SM summer camp intervention (*n* = 1). Duration of previous treatment ranged from one week to two years. Within each domain, family interview themes were reported for each identified item factor loading in the TEI (Kelley et al., 1989), as well as corresponding barriers associated with traditional therapy.

##### Acceptability

Thirteen out of 17 caregivers believed intensive summer day camp was a more acceptable SM treatment than their previous treatment. Twenty caregivers reported willingness to use strategies learned during the parent training, and 19 caregivers liked the intensive summer day camp intervention and believed the strategies taught during parent training made common sense. The one outlier preferred previous treatment because they believed CKC was more anxiety-provoking for their child than individual treatment. Seventeen caregivers reported the strategies were suitable to them, while two caregivers reported the strategies were suitable for improving speech but not anxiety, and one caregiver reported the practices of their previous therapist were more suitable. Coded themes from family interviews reveal that (a) high levels of parent satisfaction with the lead CKC clinician’s parent training sessions and (b) practice implementing skills in a community-based exposure with their child’s counselor’s support contributed to their acceptability of intensive summer day camp intervention.

Thirteen caregivers reported believing the intensive summer day camp was an accessible treatment for SM. All three caregivers whose child(ren) had no previous SM treatment believed the intensive summer day camp was an accessible form of treatment for SM, and six out of 17 caregivers believed it was more accessible than previous treatment. Four out of 17 caregivers believed the intensive summer day camp was equally accessible compared to previous treatment, and seven out of 17 caregivers preferred the accessibility of previous treatment. Two distinct themes, proximity to treatment and insurance coverage, emerged when family interviews were coded for factors contributing to accessibility. Specifically, seven caregivers from out-of-state endorsed accessibility, while five caregivers from out-of-state preferred the accessibility of previous treatment due to close proximity. Additionally, 14 caregivers endorsed partial insurance coverage.

##### Time Required

Eighteen caregivers reported satisfaction with the scheduling for the intensive summer day camp. All three caregivers whose child(ren) had no previous SM treatment believed scheduling for the intensive summer day camp was acceptable, and 12 out of 17 caregivers believed the scheduling was more acceptable than previous treatment. Three out of 17 caregivers believed the scheduling for the intensive summer day camp was similarly acceptable compared to their previous treatment, and two out of 17 caregivers preferred the scheduling of previous treatment. Work schedule flexibility, time to prepare, time of year (i.e., summer), and duration of treatment were common themes that influenced caregiver satisfaction with the scheduling of the intensive summer day camp.

Fifteen families reported the cost of the intensive summer day camp was acceptable, including all three families who had no previous treatment and twelve out of 17 families who had previous treatment. Five out of 17 families preferred the cost of their child(ren)’s previous treatment, and three out of 17 families reported cost acceptability was similar to previous treatment. The average out-of-pocket cost of the intensive summer day camp per camper was about $2747, as reported by caregivers. Camp costs $2700 per camper, but some families reported partial insurance coverage. The average cost includes all families’ out-of-pocket camp costs, travel, rental cars, lodging, food, and other miscellaneous costs (e.g., gas, dog kennel). Cost-effectiveness was the most referenced factor when caregivers reported cost satisfaction, specifically, they reported (a) similar hourly rates to traditional therapy, (b) believing their money went further at CKC compared to traditional therapy, or (c) reporting camp was less expensive than the year(s) of treatment paid for with less changes observed. Insurance coverage was the second most referenced theme for cost satisfaction.

##### Effectiveness

Eighteen caregivers reported via interview that their child reacted positively to the intensive summer day camp, whereas two caregivers reported that intensive summer day camp was harder for their child than traditional therapy. Nineteen caregivers reported camp was effective due to perceived improvements in their child(ren)’s speaking behaviors. Two of these caregivers reported that camp was effective in increasing their child(ren)’s speech, but believed their anxiety increased, too. One caregiver believed it was too early to tell if camp was effective. Common themes influencing caregiver reports about effectiveness were behavioral treatment approach and school-like treatment environment. Specifically, parents preferred the behavioral approach to other types of therapies (e.g., play). Additionally, parents commented on how the intensive summer day camp was more applicable than individual treatment because it helped their child practice speaking in a school-like environment. Eighteen caregivers reported believing their child(ren)’s counselor was competent, while two caregivers preferred their child(ren)’s more frequent or past therapist.

### 3.2. Integrity

#### 3.2.1. Counselor Integrity

All counselors (*n* = 25) self-reported daily implementation integrity ratings over 80%, with an average self-adherence rating of 97%. Similarly, counselors’ average daily ratings for adherence to the intensive summer day camp intervention was 97% (range: 83–100%), with 22 out of 25 counselors rating their implementation integrity over 95% throughout the week. Similarly, interrater agreement was 93% across daily one-hour observations of 14 counselors. No significant differences in counselor-rated integrity scores were found between (a) classes, (b) SM severity, or (c) previous treatment versus no treatment camper’s counselors.

#### 3.2.2. Caregiver Integrity

Caregivers self-reported an average of 96% (range: 60–100%) adherence during the intensive summer day camp community-based exposure activity. During the community-based exposure, caregivers completed an integrity checklist for each camper, so caregivers with two campers completed two integrity checklists. Twenty-four out of 25 caregiver-completed integrity checklists were rated at or above 80% for the community-based exposure activity, while all 25 counselors rated their camper’s caregiver’s adherence at or above 80% for the community-based exposure activity on the same scale. Inter-rater reliability between caregivers and counselors was 91%. No significant differences in caregiver-rated integrity scores were found between (a) classes, (b) SM severity, or (c) previous treatment versus no treatment.

At three-month follow-up, two out of 25 caregivers returned self-completed implementation checklists from community-based exposures completed after camp. One caregiver completed one integrity checklist with 100% adherence, while the other caregiver completed 24 checklists with an average adherence of 96.88%.

### 3.3. Effectiveness

#### 3.3.1. Daily Measures

Visual analysis for camper anxiety levels, using the counselor-rated DBRs revealed evidence of treatment effect for 14 of the 25 campers. ITSSIM results are presented as standardized mean difference, *d*. Small, medium, and large effect sizes were defined at 0.2, 0.5, and 0.8, respectively [33]. ITSSIM analysis estimated 18 campers experienced a significant (*p* < 0.05) decrease in counselor-rated anxiety with a large effect size throughout the intensive summer day camp. Specifically, 3 of 5 campers with a highest scored GAD SCARED subscale and 15 of 20 campers with a highest scored Social Anxiety SCARED subscale experienced a significant decrease in counselor-rated anxiety. Aggregated effect sizes for each class reveal improvements in counselor-rated anxiety on the DBR with large effect sizes (younger: *d* = −3.48; middle: *d* = −2.84; older: *d* = −2.04). However, six of the 25 campers experienced a significant (*p* < 0.05) increase in counselor-rated anxiety with small to large effect sizes throughout the intensive summer day camp. One camper did not experience significant counselor-rated anxiety change throughout the intensive summer day camp.

#### 3.3.2. SCARED

No significant differences in caregiver-rated anxiety scores were found at pretreatment or three-month follow-up between (a) classes, (b) SM severity, or (c) previous treatment versus no treatment; however, scores were significantly different at posttreatment for campers (c) who had previous treatment for SM versus those who had not. Campers who had received previous treatment for SM had significantly higher anxiety scores (*M* = 32.4) at posttreatment compared to campers who had not had previous treatment for SM (*M* = 18.4; *p* < 0.05). Seven campers experienced significant decreases in anxiety from pretreatment to posttreatment or posttreatment to three-month follow-up, according to the caregiver-rated SCARED. Three campers experienced a significant reduction in overall anxiety from pretreatment to posttreatment (*RCI* = −4.20 to −7.31), according to caregiver-completed SCARED. Two of these campers demonstrated a significant reduction in their highest scored subscale of anxiety (Social Anxiety) from pretreatment to posttreatment. Four out of 14 campers experienced a significant reduction in overall anxiety from posttreatment to three-month follow-up (*RCI* = −1.83 to −4.94). No campers demonstrated a significant reduction in their highest scored subscale of anxiety from posttreatment to three-month follow-up. Calculations revealed non-significant changes for campers’ anxiety from pretreatment to posttreatment (*RCI* = −0.88) and from posttreatment to three-month follow-up (*RCI* = −0.08). Aggregated DBR and SCARED results are presented in Table 3.

#### 3.3.3. SMQ

No significant differences in caregiver-rated speaking behavior scores were found between (a) classes or (c) previous treatment versus no treatment at pretreatment, posttreatment, or three-month follow-up. Differences in (b) severity scores were observed at pretreatment only, given severity was determined by pretreatment SMQ scores. Seventeen of 25 campers experienced significant improvements in caregiver-rated speaking behaviors from pretreatment to posttreatment or posttreatment to three-month follow-up. Nine campers experienced a significant improvement in caregiver-rated speaking behaviors from pretreatment to posttreatment (*RCI* = 2.4 to 24). Aggregated class effect size calculations revealed that the older class experienced significant improvements in caregiver-rated speaking behaviors from pretreatment to posttreatment (*RCI* = 3.11). Nine out of 14 in the older class experienced a significant improvement in caregiver-rated speaking behaviors from posttreatment to three-month follow-up (*RCI* = 4.8 to 18.4). Aggregated class effect size calculations revealed significant improvements in caregiver-rated camper speaking behaviors for the younger (*RCI* = 8.10) and middle (*RCI* = 7.17) class from posttreatment to three-month follow-up, and for total campers (*RCI* = 4.50).

#### 3.3.4. Speaking Behaviors

Visual analysis for camper responsive speaking behavior, using video recorded counts, revealed evidence of treatment effect for 9 of the 25 campers. Treatment effects were investigated for individual campers using ITSSIM standardized mean difference, *d*. ITSSIM analysis estimated three campers experienced a significant (*p* < 0.05) increase in responsive speech with a medium to large effect size *(d* = 0.42 to 6.11). Aggregated class data suggest small to medium effect size (*d* = 0.17 to 0.58) change for responsive speech.

Visual analysis for camper spontaneous speaking behavior, using video recorded counts, revealed evidence of treatment effect for 8 campers. Treatment effects were investigated for individual campers using ITSSIM standardized mean difference, *d*. ITSSIM analysis estimated four campers experienced a significant (*p* < 0.05) increase in spontaneous speech with a small to large effect size (*d* = 0.15 to 4.11). Aggregated class data suggest small to medium effect size (*d* = 0.13 to 0.51) change for spontaneous speech. A summary of speaking behavior change data are reported in Table 4 below.

### 3.4. Replicated Effects

As a replicated single-case AB design, it is important to assess whether changes were replicated across campers. First, replicated effects were not observed for overall acceptability on the TEQ-P due to unsatisfactory scores on the effectiveness subscale, however, the majority of caregivers endorsed adequate scores of acceptability on the quality of treatment and time required subscales. Seventeen caregivers expressed overall acceptability with the intensive summer day camp during family interviews, and almost all expressed satisfaction with the effectiveness, quality of treatment, and time required for the intensive summer day camp.

Second, replicated effects were observed for counselor-rated implementation integrity, as each counselor’s average self-ratings met or exceeded 80% adherence (*M* = 97%). Similarly, 24 out of 25 caregiver-completed implementation integrity checklists during the community-based exposure scored at or higher than 80% adherence.

Third, replicated effects were not observed on the two primary dependent variables (i.e., anxiety, speaking). However, the majority of campers (*n* = 18) experienced significant decreases in his/her highest scored subscale of anxiety (e.g., Social) throughout CKC, according to counselor-rated DBRs. Moreover, aggregated class data of caregiver-rated SCAREDs revealed non-significant improvements in overall anxiety for each class from pretreatment to posttreatment and for the younger and middle classes from posttreatment to three-month follow-up. Replicated effects across all participants with respect to improving responsive and spontaneous speaking behavior, were not observed. Only three campers improved responsive speaking and four improved spontaneous speaking from the start to end of camp. However, caregivers endorsed significant speaking behavior improvements on the SMQ for the older class at posttreatment and for the younger and middle classes at three-month follow-up.

## 4. Discussion

### 4.1. Study Contributions to the Literature

The current study contributes to current literature by investigating the acceptability, integrity, and effectiveness of a 5-consecutive day intensive summer day camp for children with SM (i.e., CKC) through a replicated single-case AB design. This study extends on prior research about SM by exploring behavioral therapy implemented within an intensive format. While literature is well-established to suggest behavioral therapy is the most effective treatment for SM, it has not been evaluated in an intensive intervention format before. However, there is research available about intensive interventions for similar disorders (e.g., social phobia, OCD) to suggest initial effectiveness findings. This study builds in additional methodological rigor to previous studies about intensive intervention by thoroughly investigating the acceptability and integrity of intensive intervention. Given the rarity of the SM diagnoses, this sample size of 25 provides a robust beginning to research on intensive interventions for this anxiety disorder subtype. The examination of both individual change and replicated change further highlights the necessity of a combined approach when working with youth with SM who present unique symptom profiles.

### 4.2. Treatment Acceptability

Caregivers inconsistently reported the intensive summer day camp as an acceptable treatment approach for SM. Specifically, only six caregivers endorsed satisfactory scores (110 or higher) [34] for the intensive summer day camp’s overall acceptability on the TEQ-P. This overall score was impacted by inadequate scores of camp effectiveness on the TEQ-P, as only six caregivers endorsed acceptable scores (36 or higher) [34]; however, the majority of caregivers endorsed satisfactory scores for the time required (*n* = 16) and quality of treatment (*n* = 17) of the intensive summer day camp on the TEQ-P. Conversely, family interviews revealed very high levels of caregiver-reported satisfaction with the intensive summer day camp, including for overall acceptability, effectiveness, time required, and quality of treatment.

Acceptability results from this study differ from previous intensive intervention literature for similar disorders, in which high levels of caregiver satisfaction were endorsed consistently [8,15,18]. There are important potential explanations for why differences in acceptability exist between this study and past intensive intervention literature for similar disorders. First, it is possible caregivers from this study were less likely to endorse acceptable rates of effectiveness on the TEQ-P at posttreatment because they had not yet observed change in their child’s speech at school, which is both a naturalistic exposure and the primary environment where SM is reinforced [4]. This is an explanation unique to SM that may influence why differences in acceptability existed between this study and previous intensive intervention literature. This explanation is supported by evidence from this study that counselors were more likely than caregivers to recognize the breakdown in the SM avoidance cycle (i.e., effectiveness changes) by the last day of camp, as they endorsed significant improvements in anxiety throughout camp for majority of campers (*n* = 18). Caregivers from this study endorsed significant improvements in speaking behaviors (i.e., effectiveness) for majority of campers (*n* = 17), but not until three-month follow-up. Second, this study differed from previous intensive intervention literature because it was the first to use a reliable survey to assess satisfaction with intensive intervention effectiveness. There are nine items on the TEQ-P that measure effectiveness, which is more than the overall acceptability measure used in the most comparable study [8]. It is possible acceptability surveys and interviews from other studies assessed more for time required and quality of treatment than effectiveness, and that their results reflect this.

Despite differences in acceptability scores between this study and previous literature, there were many overlaps that add to the intensive intervention literature. Parent scaffolding of behavioral strategies (e.g., community-based exposure) and parent training, were the two distinct themes positively contributing to caregiver acceptability in this study’s family interviews consistent with prior literature [18]. These identical reports align with the use of BICs as disruptive innovations, as their purpose is to treat the disorder, while relaying all necessary behavioral skills to caregivers, during a condensed treatment approach [11]. The majority of caregivers in this study reported cost acceptability compared to cost of previous treatments, despite an average per camper out-of-pocket cost over $2700 and inadequate effectiveness scores on the TEQ-P. Consistent with findings from two prior studies [8,18], it is possible intensive interventions are most accessible to higher income families, given travel and therapy costs to access experts for less prevalent disorders (e.g., SM). These data provoke critical discussion about whether intensive interventions are a disruptive innovation to behavioral therapy in its traditional format. While the intensive format is disruptive, these cost data suggest intensive intervention may not address cost barriers to treatment as originally hypothesized. Third, as expected, almost all caregivers reported satisfaction with this study’s scheduling and counselor implementation competency. This is consistent with prior report [11] of BICs as disruptive innovations to circumvent traditional behavior therapy barriers, such as scheduling conflict for weekly therapy and lack of access to expert clinicians.

### 4.3. Treatment Integrity

As anticipated, all camp counselors self-reported excellent daily implementation integrity ratings (*M* = 97%), exceeding the 80% adherence threshold recommended [23]. This finding aligns with previous IBTSM research that behavioral therapy for SM can be implemented with high rates of integrity [34], *M* = 99% and supports findings from prior intensive intervention literature for similar disorders, e.g., [16]. As hypothesized, counselor integrity was protected by consistency in counselor-camper match throughout treatment, brevity of treatment, counselor competency, and detailed treatment protocols [22]. Interrater agreement between counselor self-ratings and a licensed clinician was 93% across daily one-hour observations of 14 counselors, which exceeded the average 81% agreement rate recommended [24]. This is promising data for intensive interventions as disruptive innovations aim to address the barrier of implementation competency associated with low-prevalence disorders. Specifically, novice counselors in related professional fields (e.g., psychology graduate students, social workers) displayed superb treatment adherence to behavioral therapy following a one-day training from a SM expert clinician.

Similarly, caregivers’ average self-rated implementation integrity during the intensive summer day camp, during which they led a community-based exposure activity for their child, was 96%, which exceeded the 80% standard [23]. No previous study has reported caregiver implementation integrity for intensive interventions. All counselors scored their camper’s caregiver’s implementation at or above 80% integrity during the community-based exposure, and the rate of interrater agreement (91%) was higher than the recommended threshold of 81% [24]. As hypothesized, it is likely caregiver integrity was protected by similar factors described above for counselors. High levels of caregiver treatment adherence is important for the intensive intervention literature for SM. Ultimately, these adherence rates imply caregivers obtained excellent behavioral therapy skills for SM as a result of daily two-hour parent trainings from a SM expert clinician. High rates of caregiver adherence suggest caregivers perceived the behavioral treatment approach as highly acceptable [14], which is impressive given the heterogeneous sample for this study, which included caregivers whose children had no previous treatment for SM, previous behavioral treatment, and previous play therapy.

These acceptability and integrity data provide initial support for intensive interventions as disruptive innovations to address the barrier of access to expert clinicians for a low-prevalence disorder like SM. Specifically, they provide initial support for the component of BIC interventions to access an expert briefly and then generalize skills learned to home, school, and public [11]. It was surprising, then, that only two caregivers showed evidence of tracking implementation integrity between intensive summer day camp and three-month follow-up, especially in light of the ability to receive effectiveness data from the majority of families at three-month follow-up (reported below). It is likely the integrity rating sheets were burdensome to track and subsequently turn in via email scanning/mail. Though there is information to be desired about caregiver implementation between posttreatment and three-month follow-up, it can be hypothesized caregivers continued to implement behavioral therapy strategies successfully. This hypothesis is supported by the link between implementation integrity and intervention effectiveness discussed further below.

### 4.4. Anxiety Symptoms

This was the first intensive intervention study for an anxiety disorder to assess within-intervention subtype anxiety changes, and the results are promising. Counselors indicated that 18 of 25 campers experienced a significant decrease in their highest subtype of anxiety during camp. The younger, middle, and older camp classes experienced anxiety change on counselor-rated DBRs with large effect sizes. In contrast to significant counselor-rated changes in anxiety for those participating in the intensive summer camp, only three campers’ caregivers endorsed significant decreases in anxious symptoms from pretreatment to posttreatment. These data differ from a previous intensive intervention study that found all five participants experienced a significant decrease in separation anxiety disorder symptoms at posttreatment [20].

These outcome data pertaining to anxiety symptoms are important, as they align with the behavioral conceptualization of the communication hierarchy associated with SM. Specifically, counselors increase exposure to speech (e.g., forced-choice questions vs. open-ended questions) at an appropriate pace to promote speech, and the pace of this treatment hierarchy was individualized to each camper. Counselor DBR data suggest some change was occurring for most children’s highest rated anxiety during camp, which aligns with the behavioral mechanism of change theory that lower rates of anxiety promote success with exposure to speech in incrementally challenging situations. This process is identical to the goal of behavioral treatment for SM, which is to decrease symptoms of SM as children become more confident with increasingly challenging exposures. Results from this study suggest counselors, or providers of the intensive intervention, are in a better position to identify those symptoms changes during treatment than caregivers who were not present throughout the day.

### 4.5. Speaking Behaviors

Nine of the 25 campers experienced significant improvements in caregiver-rated speaking behaviors from pretreatment to posttreatment, as reported on the SMQ. This increased within other campers at follow-up (9 out of 14), suggesting a lag effect of speech behavior. These data are consistent with the behavioral mechanism of change theory that anxiety was reduced prior to seeing changes in speech. Specifically, speech was established in a safe environment (i.e., camp) with repetitive exposures before it was generalized to school/community with improvements noticeable to caregivers.

Study results also support the importance of looking at treatment outcomes, developmentally. For example, aggregated class effect size calculations revealed that the older class experienced significant improvements in caregiver-rated speaking behaviors from pretreatment to posttreatment (*RCI* = 3.11). Yet, those changes appeared to max out for the older group, as speaking behavior was maintained at three-month follow-up. Aggregated class effect size calculations revealed significant improvements in caregiver-rated camper speaking behaviors for the younger (*RCI* = 8.10) and middle (*RCI* = 7.17) class from posttreatment to three-month follow-up, and for total campers (*RCI* = 4.50). Similar developmental differences in treatment gains at one-year follow-up after six months of behavioral treatment for SM (*n* = 24) were found compared to prior studies [18] which found that 78% of three to five-year-olds did not meet diagnostic criteria for SM, compared to 33% of six to nine-year-olds. While the present study did not assess for diagnostic criteria, speaking behaviors are the most related SM diagnostic symptom, and this study provides more evidence to support that breaking the speech avoidance cycle [4] can result in better outcomes [18].

This was the first study to track daily child speaking behaviors and code words per minute during a 5-consecutive day intensive summer day camp for SM. This study found that three campers improved responsive speech throughout camp and that four campers improved spontaneous speech over camp. This methodology of measuring speech raises the bar from previous SM literature [35,36]. Both articles reported increases in response rate to over 50% of prompted opportunities by halfway through treatment. Campers do not escape responses during camp (i.e., response rate of ~100%), and are thus inundated with learning opportunities that speaking cannot be avoided. It makes sense, then, that campers were likely to generalize this learning to different contexts (i.e., school and public) after camp was over via parent SMQ ratings at three-month follow-up. As a result, evidence from this study demonstrated caregiver-rated camper speaking behaviors improved more between posttreatment and three-month follow-up than pretreatment to posttreatment. Verifying those changes within the school context will be essential in future research.

### 4.6. Limitations

The present study is limited by reliability and validity of baseline measures. First, consistency in baseline anxiety levels could not be established due to significant differences in caregiver-rated SCARED scores from intake to pretreatment. Second, due to intensive summer day camp structure, there was not ample time/opportunity to establish a baseline of speech behavior before intervention; however, ITSSIM software was chosen specifically for analysis because it addresses this clinical practice limitation. Third, the DBR was a truncated scale from zero to two, which could have affected how changes were detected, most notably resulting in large effect sizes. Additionally, the lack of a comparison group is a limitation of this study. Without a comparison group, it is impossible to know if camper changes over time were due to intervention or not. Maturation is a possible explanation for changes over time and is a threat to the internal validity of this study without a comparison group. Missing data from the three-month follow-up is a limitation of the study. It is also possible a biased subsample responded to study recruitment materials.

### 4.7. Implications for Research

This pilot feasibility study sets a foundation for the investigation of intensive summer day camps as a treatment for SM, as to date, intensives have only been reported in the literature for other anxiety-related conditions [8,20]. Replications of this study, with refinements, are warranted to address its limitations. First, more research is needed to better understand family acceptability of intensive summer day camp intervention as a treatment for SM to clarify inconsistencies in this study’s results of effectiveness acceptability. Future research should consider using the TEQ-P to assess family acceptability, but place it at a follow-up data collection time point for more accurate perceptions about intervention effectiveness. Additionally, a future study could explore the psychometric properties of the semi-structured interview used in this study, as it could be an important tool for quality improvement efforts. Continued research efforts to evaluate the acceptability of intensive summer day camps as a treatment for SM are critical to establishing its feasibility.

More research is needed to better understand factors contributing to implementation integrity of behavioral therapy during and after the intensive summer day camp for children with SM. Given the high rates of integrity by counselors and caregivers during camp after only a brief training from the lead CKC clinician, it would be interesting to assess which parts of the parent and counselor training promote integrity. Most notably, more research is needed to understand parent implementation integrity of SM behavioral therapy after intensive summer day camp is over. Specifically, understanding which components of acceptability optimize the likelihood parents will practice SM behavioral therapy and track their implementation integrity would be useful for practice.

Future research may address limitations from this study’s evaluation of the effectiveness of intensive summer day camp intervention. Researching the possible treatment effects of the lead-in session is warranted. As such, ensuring a baseline for speaking behaviors and anxiety levels is established before intervention will result in more valid data. Additionally, implementation with more severe SM campers and with a control group will make for exciting progress in the literature. Next, a better understanding of responsive speech changes to spontaneous speech will be warranted for the intensive summer day camp intervention research. Additionally, controlling for a clinic effect is essential in future research. While this study’s intensive intervention simulated a school-like environment, it is essential to transition treatment to the child’s unique environment, especially school.

### 4.8. Implications for Practice

This study informs initial implications for practice. First, this study suggests SM behavioral therapy can be implemented with integrity by novice clinicians and parents after training from a SM expert clinician. Given training was brief and there is a need for SM-competent clinicians, SM expert clinicians may be motivated to provide training to clinicians, schools, and parents. Second, a lack of replicated change across campers at posttreatment and inadequate follow-up response rate may reveal a need for more frequent booster sessions. This implication for practice would be interesting to investigate alongside increasing access to teletherapy, as it is also perceived as a cost-effective and accessible treatment for anxiety disorders [37].

## Figures and Tables

**Figure 1 children-09-01732-f001:**
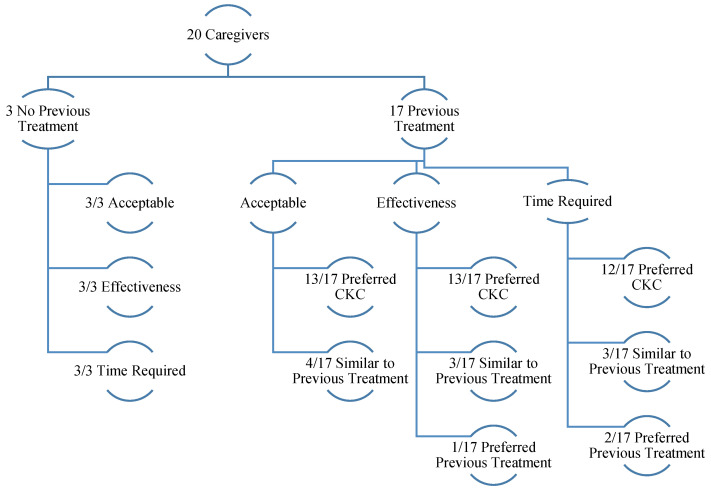
Family interview summary.

**Table 1 children-09-01732-t001:** Confident Kids Camp Daily Activities.

Time of Day	Activities
Morning Activities	Free play Circle time or relaxation
Mid-morning	Psychoeducation about why we practice Exposure activity (e.g., police officer, princess visit, show and tell) Set goals for afternoon exposure Snack and recess Lunch
Mid-afternoon	Field-trip activity (e.g., creature conservancy, art project) Recess Classroom-based practice for interrupting, giving compliments, Person Bingo
Afternoon	Prize store

**Table 2 children-09-01732-t002:** TEQ-P Results (*N* = 25).

Class	Overall Acceptability	Treatment Quality	Time Required	Effectiveness
Younger	96.00 (15.66)	60.57 (7.72) *	9.00 (3.05) *	26.43 (8.52)
Middle	99.89 (12.56)	59.56 (4.69) *	10.00 (2.00) *	30.33 (4.12)
Older	98.22 (8.89)	57.55 (4.77) *	9.33 (1.73) *	31.33 (4.12)
Total	98.2 (11.95)	59.10 (5.40) *	9.48 (2.20) *	29.60 (6.73)

* Indicates a high level of acceptability based on established interpretation criteria [21].

**Table 3 children-09-01732-t003:** Aggregated Anxiety Data.

**Daily DBR: Range 0–2**							
**Class**	**Baseline**	**Monday**	**Tuesday**	**Wednesday**	**Thursday**	**Friday**	** *d* **
Younger	1.83	1.27	1.00	1.09	1.28	0.88	−3.48 *
Middle	1.63	1.08	0.95	0.94	0.71	0.79	−2.84 *
Older	1.66	1.46	1.29	1.04	0.98	0.89	−2.04 *
Total	1.71	1.27	1.08	1.02	0.99	0.85	−2.73 *
**Parent SCARED: M (SD)**							
**Class**	**Intake**	**Pre**		**Post**	**RCI**	**3-month Follow-up**	**RCI**
Younger	24.42 (7.98)	31.57 (13.33)		31.43 (15.09)	−0.03	24 (12.62)	−1.36
Middle	35.44 (15.95)	37.11 (15.74)		32.89 (14.56)	−0.77	31 (15.95)	−0.35
Older	27.44 (10.73)	33.89 (13.94)		24.89 (10.39)	−1.65	31.4 (11.41)	1.19
Total	29.48 (12.70)	34.40 (14.04)		29.60 (13.30)	−0.88	29.14 (12.91)	−0.08
**Child SCARED: M (SD)**							
	**Intake**	**Pre**		**Post**	**RCI**	**3-month Follow-up**	**RCI**
Total	28.73 (16.76)	25.91 (13.39)		23.10 (12.87)	−0.51	Not Reported	

* = statistically significant reductions in anxiety observed based on small, medium, and large effect sizes of 0.2, 0.5, and 0.8, respectively [33].

**Table 4 children-09-01732-t004:** Aggregated Speaking Behavior Data as Measured by Parent SMQ Ratings.

**Selective Mutism Questionnaire (SMQ)*:* M (SD)**							
**Class**	**Intake**	**Pre**		**Post**	**RCI**	**3-Month Follow-Up**	**RCI**
Younger	16.14 (7.80)	22.43 (9.22)		24.14 (8.36)	1.37	34.25 (5.44)	8.10 *
Middle	18.78 (7.46)	21 (9.22)		21.44 (10.86)	0.35	30.4 (10.09)	7.17 *
Older	19.89 (5.33)	21.67 (6.20)		25.56 (9.59)	3.11 *	26 (5.87)	0.35
Total	18.44 (6.75)	21.64 (7.92)		23.68 (9.52)	1.63	29.93 (7.78)	4.50 *
**Daily Speaking Behaviors: Words per Minute**							
**Class**	**Baseline**	**Monday**	**Tuesday**	**Wednesday**	**Thursday**	**Friday**	** *d* **
**Responsive Speech**							
Younger	2.78	2.13	2.99	3.04	3.63	3.75	0.19
Middle	1.43	1.71	2.67	3.08	2.57	4.07	0.58
Older	1.33	0.94	2.67	2.96	1.74	2.00	0.17
Total	1.85	1.59	2.78	3.03	2.65	3.27	0.32
**Spontaneous Speech**							
Younger	0.36	0.19	0.56	0.67	0.98	1.59	0.13
Middle	0.32	0.34	0.10	0.51	2.03	2.23	0.51
Older	0.59	0.18	0.37	1.36	3.78	4.27	0.44
Total	0.42	0.24	0.34	0.93	2.26	2.70	0.29

* = significant improvements observed (RCI > 1.8).

## Data Availability

Data supporting reporting results are held by the first author.

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
