# Peer review of "A Pilot Feasibility Study of an Intensive Summer Day Camp Intervention for Children with Selective Mutism"

_children, 2022, doi:10.3390/children9111732_

Round 1

Reviewer 1 Report

Dear authors, thank you very much for the opportunity to read your paper. I have only one but important recommendation. Your research sample is very small. For this reason, I would avoid using percentages. The data can be interpreted more like in qualitative research, as percentages (rates) are misleading in such small numbers.

Author Response

Reviewer 1: Your research sample is very small. For this reason, I would avoid using percentages. The data can be interpreted more like in qualitative research, as percentages (rates) are misleading in such small numbers.

Response to Reviewer 1 Feedback: We have made these changes throughout the manuscript and greatly appreciate this feedback as we fully agree that the use of percentages when working with small sample sizes can lead to misleading interpretations.

Reviewer 2 Report

Dear authors,

It is good research to promote the social inclusion of children with different abilities. A good theoretical foundation, adequate work methodology and coherent discussion were established. Authors are encouraged to materialize the research projection.

Some considerations.

1. Define abbreviation “AB” (line 14).

2. It is strange that a statistical analysis section has not been included. It should be included to describe the type of analysis and programs used (it should be written at the end of the "materials and methods" section).

3. It should be indicated in table 2 if there were significant differences between the subscales.

4. Statistical significance asterisks are missing in Table 3.

5. Do the authors see any strength of the study? Using validated instruments is a strength. What do you think? (section “limitations”). Some research strengths should be included.

6. Complete with information on the ethics committee that approved the study, consents for parents or guardians, and assents for children (“Institutional Review Board Statement” section).

Thanks.

Author Response

Thank you for the very helpful feedback. Please see the attached cover letter for a summary of how your feedback helped to improve our manuscript. 
